# Enhanced Effect of Botulinum Toxin A Injections into the Extensor Digitorum Brevis Muscle after Local Mechanical Leg Vibration: A Case Report

**DOI:** 10.3390/toxins13060423

**Published:** 2021-06-15

**Authors:** Harald Hefter, Judith Beek, Dietmar Rosenthal, Sara Samadzadeh

**Affiliations:** 1Department of Neurology, University of Düsseldorf, Moorenstrasse 5, D-40225 Düsseldorf, Germany; Judith.beek@gmx.de (J.B.); Dietmar.Rosenthal@med.uni-duesseldorf.de (D.R.); sara.samadzadeh@yahoo.com (S.S.); 2Department of Pediatrics, Burgerstrasse 211, D-42859 Remscheid, Germany

**Keywords:** local mechanical leg vibration, vibration ergometry training, efficacy of botulinum toxin therapy, extensor digitorum brevis muscle, improvement of muscle action potentials

## Abstract

Background: The aim of this study was to demonstrate an increase in muscle action potentials and an enhancement of the efficacy of botulinum toxin (BoNT) after mechanical leg vibration. Methods: A 53-year-old healthy male volunteer underwent vibration ergometry training (VET) every morning and every evening for 10 min for 14 days. Compound muscle action potential (CMAP) of the right (R) and left (L) extensor digitorum brevis (EDB) muscle was analyzed by supramaximal peroneal nerve stimulation before and after VET 12 times during the 14 days. Thereafter, VET was stopped and 20 U incobotulinumtoxin (incoBoNT/A) were injected into the right EDB. During the following 10 days, CMAP of both EDBs was tested 12 times. Results: Under VET, the CMAP of both EDBs significantly increased (L: *p* < 0.01; R: *p* < 0.01). During the first 14 days, CMAP of the left EDB before VET was significantly (<0.008) lower than 20 min later after VET. This was not the case for the better trained right EDB. After day 14, CMAP of the untreated left EDB further increased for 6 days and then decreased again. In the right EDB, BoNT-treated EDB CMAP rapidly and highly significantly (*p* < 0.0001) decreased during the first 48 h by about 90%, from a level of about 14 mV down to a plateau of around 1.5 mV. Conclusion: Local mechanical leg vibration has a short- and long-term training effect. Compared to other studies analyzing the reduction in EDB CMAPs after BoNT injections, the reduction of EDB CMAPs in the present study observed after combined application of BoNT and VET was much faster and more pronounced.

## 1. Introduction

Injections of botulinum neurotoxin (BoNT) are used for symptomatic treatment of a wide spectrum of disease entities [1]. In a variety of indications, repetitive injections have to be performed to achieve a permanent level of improvement [2]. This bears the risk of antibody (AB) formation against the 750 kDa botulinum neurotoxin complex, not only against the hemagglutinin and non-hemiagglutinin complex proteins but also against the 150 kDa large BoNT molecule itself [3]. Some of these antibodies reduce or even neutralize BoNT action (NABs [4,5]). If possible, their induction should be avoided in the course of BoNT treatment. High doses per session and duration of treatment are the main risk factors for NAB formation [3,6,7]. Since repeated BoNT injections are necessary for a good long-term outcome, reduction in dose per session in BoNT long-term therapy has been emphasized [3,6,7]. Therefore, methods which enhance the efficacy of BoNT and allow a reduction in BoNT dose per session are very much appreciated.

Several guidance techniques (computer tomography, ultrasound, electromyography) have been developed to apply BoNT precisely. These may be helpful to reduce BoNT dose per structure and session [8]. In the present research, we focus on mechanical leg vibration.

Vibration is known to influence muscle strength and the growth of small vessels [9,10]. Little is known about the interaction between BoNT and vibration and their combined clinical application. For cosmetic application, vibration is used to reduce the pain of BoNT injections, but not to increase the efficacy of BoNT injections [11,12]. In patients with rotational cervical dystonia, injections of BoNT/A into the affected sternocleidomastoideus muscle significantly reduce vibration-induced facilitation of motor evoked potentials after 6 weeks [13]. In patients with multiple sclerosis, local muscle vibration, as well as BoNT injections, reduced spasticity after 10 weeks. A combination of vibration and BoNT did not appear to be superior to vibration or BoNT alone [14].

Mechanical leg vibration is a unique technique originally developed for bicycle professionals to enhance their training during winter [15]. However, one has to be cautious when transferring information on mechanical leg vibration from athletes to normal subjects and to patients. Especially before clinical application, some parameters of vibration ergometric training (VET) have to be studied in controls. Thus far, there has been no research on mechanical leg vibration and the effect of BoNT. Therefore, the present pilot and case study was performed.

## 2. Methods

The local ethics committee of the University of Düsseldorf approved an application of our team on optimization of therapy and determination of antibodies in BoNT treatment (number: 4085).

### 2.1. Description of the Vibration Ergometry Device

In the present study, the first prototype of a vibration ergometer was used. When studying sports medicine, Dieter Quarz developed the idea of improving the winter training of cycling professionals by combining ergometry and vibration. He built a prototype (see Figure 1) by decoupling the crank from the excenter driven bottom bracket and the pedals, which allowed a more intensive training of the legs and the avoidance of entire body vibrations, which inevitably occur during the use of the usual vibration platforms, with the risk of central nervous system damage [16]. In the present study, a vibration frequency of 15 Hz, a vibration amplitude of 0.4 cm and a low power of 20 Watts was used. This choice of parameters was made to test the parameters of VET under which handicapped patients were also able to perform VET.

### 2.2. Subject

A 53-year-old male right-handed healthy volunteer gave written informed consent after receiving information on the purpose of the study. He had a history of a fracture of the left ankle without nerve injury during training for ski-jumping at the age of 9 years. From the age of 18 until the age of 50, he had performed long distance running (>10 miles/day). Clinical neurological examination was normal. Peroneal nerve conduction velocities were within normal limits, and the CMAPs of the right EDB were larger than those of the left EDB.

### 2.3. Design of the Study

EDB CMAPs were determined in a routine EMG laboratory (EMG-lab) of the Neurological Clinic of the University Hospital in Düsseldorf (Germany) at 8 a.m. For the determination of the EDB CMAPs, the volunteer had to lie either on the left or the right side. In the first step, the optimal localization of surface electrodes over the belly of the EDB was found by trial and error. This localization was marked with waterproof ink so that the electrodes could be taken off and replaced easily without causing variability in the EDB CMAP of more than 10%. EDB CMAPs were determined as an average from 5 consecutive supramaximal 1 Hz peroneal nerve stimulations. EDB CMAP determination was always performed on both sides and lasted about 5 min per side.

Thereafter, cables of the surface electrodes were removed, but electrodes remained in place. The volunteer went to the nearby laboratory where a technician had already started the device to perform vibration ergometry training (VET). VET was performed for 5 min. Thereafter, while sitting on the VET device, the volunteer had to insert a pause of 5 min. Then, a second VET had to be performed for a 5 further min. After VET, the volunteer went back to the EMG-lab for a second determination of EDB CMAPs. Time from the onset of the first to the end of the second EDB CMAP measurement was less than 50 min.

At 6 p.m., the entire procedure was performed a second time. During the first 14 days, VET was performed every day even during the weekend. EDB CMAPs were determined 12 times during the 2 weeks.

On day 14 after EDB CMAP determination and VET in the morning, a total dose of 20 U incobotulinumtoxin A (incoBoNT/A; Xeomin^®^) in a dilution of 100 U/2 mL (=20 U/0.4 mL) were injected in equal portions at 5 different sites into the right EDB muscle. On the evening of day 14, EDB CMAP determination was performed without VET for the first time. Between the first and the second measurement of EDB CMAP, the volunteer lay on his back in a completely relaxed position for 20 min without any foot movements. During the next 10 days, a further 12 paired EDB CMAP measurements were performed, always with a pause of 20 min.

### 2.4. Statistics

A two-group (before/after) repeated measurement ANOVA (rmANOVA) was performed to analyze EDB CMAPs before and after VET and during the first 14 days and to compare EDB CMAPs before and after a 20 min pause without VET. Parametric (Pearson correlation) and non-parametric (Spearman’s rho) statistics were calculated between the duration of VET and EDB CMAPs. All statistical procedures were performed with the SPSS^®^ statistics package (version 25; IBM, Armonk, NY, USA).

## 3. Results

### 3.1. Improvement of EDB CMAP after 10 min Vibration Ergometry Training

Nerve conduction velocities of the peroneal nerves of the volunteer were within normal limits (right side: 52 m/s; left side: 48 m/s). The left EDB CMAPs were significantly (*p* < 0.001) lower than the right EDM CMAPs (Table 1). The rmANOVA revealed a significant (*p* < 0.008) improvement of the left EDB CMAP after VET for 10 min during the first two weeks, but not for the right EDB CMAP (Table 1).

From days 14 to 24 after cessation of VET, CMAPs of both EDBs were slightly smaller when they were measured a second time after a pause of 20 min (Figure 2). For the right EDB, this difference was significant (non-parametric testing; *p* < 0.02; Table 1).

### 3.2. Improvement of EDB CMAP with Vibration Ergometry Training over 14 Days

For both EDBs, CMAPs increased with the duration of VET (Figure 2). This increase of the left CMAP was steeper and significant before (*p* < 0.05) and after (*p* < 0.01) VET. For the right EDB, this increase was significant (*p* < 0.01) only after VET (Table 2). The regression lines between CMAP and the duration of VET for CMAP measurements before and after VET were parallel for both EDBs during the first 2 weeks (Figure 1).

### 3.3. Development of Left EDB CMAP after Cessation of VET

After day 14, the CMAP of the left EDB continued to increase for a further 6 days. Thereafter, left EDB CMAP started to decrease again. After 10 days without VET, the left EDB CMAP was still much higher than before VET 24 days previously. No significant difference was found between CMAPs before and after a rest of 20 min (Table 1).

### 3.4. Development of Right EDB CMAP after Cessation of VET and BoNT Treatment

By day 14, about 9.6 h after injection of 20 U incoBoNT/A, the CMAP of the right EDB had declined by more than 56% (=6.83/12.17) and 47.7% (=7.00/14.67) 20 min later. The next day, around the time of BoNT/A injection, the right EDB CMAP was reduced to 3.33 mV (=27.4% = 3.33/12.17) and to 18% (=2.67/14.67) 20 min later. To our knowledge, this is the fastest reduction in CMAP after BoNT ever reported. After 48 h, CMAP approached a flat plateau of about 1.5 mV which did not change during the next 8 days.

## 4. Discussion

In the present study, for the first time, an interaction between vibration ergometry and application of botulinum toxin is demonstrated. A short-term effect, as well as an even larger long-term effect, of VET is shown. Furthermore, the efficacy of BoNT injections had been enhanced by VET. These three aspects will be discussed in the following section.

### 4.1. The Immediate or Short-Term Effect Of Vibration Ergometry

The comparison of EDB CMAPs before and after 10 min VET revealed a significant improvement in the less trained left EDB and a tendency to higher values in the more trained right EDB (Table 1). This is consistent with the training’s effect after vibration observed in other healthy volunteers and cycling professionals [17,18,19]. The interpretation of this effect is that vibration activates otherwise silent endplate contacts. As soon as VET was stopped at day 14, this short-term activation disappeared from days 14 to 24. Repetition of EDB CMAP measurements no longer revealed an increase, but rather a decrease in CMAP after a pause of 20 min, which was significant for the right, but not for the left, EDB (Table 1).

### 4.2. The Long-Term Effect of Vibration Ergometry Training

The use of vibration ergometry training (VET) has already been analyzed in healthy volunteers and athletes at the University of Sports Medicine in Cologne (Germany). An increase in muscle strength, increase in strength endurance and blood flow and growth of blood vessels could be demonstrated [17,18,19]. When repeated VET was applied to the healthy volunteer in the present study, CMAP increased by 0.2524 mV/day from 6.5 mV to 10 mV in the less trained left EDB and by 0.2207 mV /day from 10.9 mV to 14.1 mV in the right EDB. After 14 days of VET, CMAP was well above the mean values observed in the literature (e.g., mean value: 8.3, S.D.: 3.1 in [20]). Even after cessation of VET, this increase of the left EDB CMAP continued and reached a peak value close to 12 mV. During the next 4 days, CMAP started to decline. This increase in CMAP during the 14 days of VET is consistent with the results from cycling with and without vibration demonstrating the growth of vessels and muscle fibers during VET.

### 4.3. Enhancement of Efficacy of BoNT by Vibration Ergometry Training

It has been demonstrated in cell-based experiments [20,21,22] and electrical stimulation in patients with spasticity or other movement disorders [23,24,25] that the higher the exocytosis and turnover at the endplate is, the better is the uptake of BoNT. We therefore interpreted the significant (*p* < 0.05) short-term effect with an increase in mean EDB CMAP of more than 0.9 mV immediately after VET as an endplate activation and a promising sign for BoNT/A application.

Indeed, the efficacy of incoBoNT/A after conditioning VET over 14 days was striking. A rapid and large decline in right EDB CMAP after the injection of 20 incoBonT/A of more than 50 % in the first 10 h and of about 75% in the first 24 h could be measured. After 48 h, the maximal reduction of more than 90% was observed. This has not been described or observed to date, although development of EDB CMAP has been intensively studied under various conditions [26,27,28,29].

Analysis of EDB CMAP reduction has been carried out after injection of up to 200 U abobotulinumtoxin A (aboBoNT/A Dysport^®^) [27,28], after injection of up to 32 U onabotulinumtoxin A (onaBoNT/A; Botox^®^) [26,28,29] and after injection of up to 32 U incoBoNT/A [29]. There is a long debate on the conversion ratios between the three different BoNT/A preparations that are licensed in Europe [30]. For the EDB muscle, the conversion ratios have been intensively analyzed [28,29]. Under the assumption of a conversion ratio of 1:1 between inco- and onaBoNT/A and 1.57–3:1 between ona- and aboBoNT/A [28,29], the equivalent doses for incoBoNT/A range from 2 to 66 U. Thus, the 20 U incoBoNT/A used in the present study did not exceed the dose ranges used in previous studies on EDB CMAP reduction [26,27,28,29]. Therefore, it is unlikely that the rapid and pronounced decline in EDB CMAP in the present study occurred only because of the use of a rather high dose of BoNT/A.

The onset of the decline in EDB CMAP within the first 48 h has been mentioned previously [26], but in none of the previous studies was such a rapid decline of EDB CMAP observed as seen in the present study. Usually, the peak effect was reached between 7 and 21 days [26,27,28,29].

Furthermore, after VET, EDB CMAP was higher than 12 mV in the left and 14 mV in the right EDB which is larger than the mean EDB plus 1 standard deviation in most of the previous studies [26,27,28,29]. Forty-eight hours after injection of 20 U of incoBoNT/A in the present study, CMAP dropped down to values well below the maximally reduced mean EDB CMAP (e.g., mean value: 3.0 mV, S.D.: 0.9 mV in [26]) minus 1 standard deviation. This emphasizes not only the rapid but also extremely effective response to incoBoNT/A in the present study.

We, therefore, think that the highly significant EDB CMAP reduction observed after injection of 20 U incoBoNT/A in the present study is not due to the use of an excessively high dose of BoNT/A but due to the previous vibration ergometry training which had ideally prepared the endplates of the EDB for the uptake of BoNT/A.

## 5. Conclusions

Vibration ergometry training has a short-term and a long-term positive effect on the strength of lower leg muscles in healthy controls. This may turn out to be helpful for patients with lower leg weakness. The combined application of conditioning local mechanical leg vibration and BoNT/A injections led to a pronounced efficacy of BoNT/A. If confirmed, this finding will have a variety of clinical implications for botulinum toxin management. The injection cycle duration may be increased, the dose per session decreased and the risk of antibody formation reduced. Therefore, further well-designed studies are recommended to compare the efficacy of BoNT/A with and without VET, controlling for the duration and intensity of vibration, the dose of BoNT/A and the outcome.

## 6. Strength and Limitations of the Present Study

The strength of this case study is the clear-cut demonstration of a short-term and long-term effect of VET as well as a pronounced effect of BoNT action after VET. The original design of this preliminary study was intended to include five healthy volunteers. However, we did not succeed in finding further volunteers who had time for repeated VET training in the morning and evening every day including during the weekend over two weeks and for an experimental BoNT/A injection and an extension phase of the study with EDB measurements in the morning and evening. Compared to other publications, EDB measurements of the single subject in the present study were representative. However, of course, to study the influence of VET on the efficacy of BoNT, further studies with more subjects are necessary. However, the main purpose, to prepare the design of a first clinical application of VET in BoNT/A-treated patients, was achieved.

## Figures and Tables

**Figure 1 toxins-13-00423-f001:**
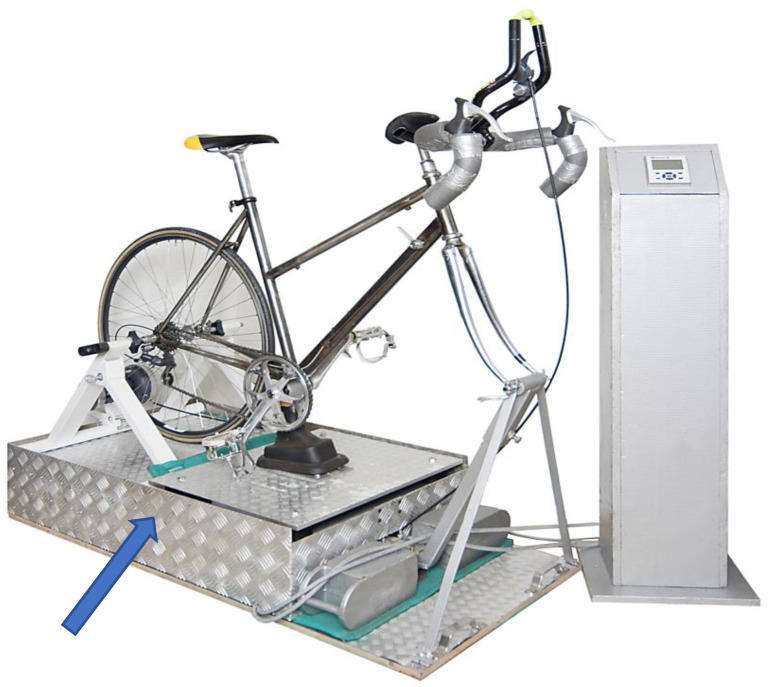
The prototype of a vibration ergometer device invented by Dieter Quarz. This photo is presented with his permission. The arrow in the figure shows the vibration platform, which vibrates the bottom bracket and is driven by a strong electromotor with excenter.

**Figure 2 toxins-13-00423-f002:**
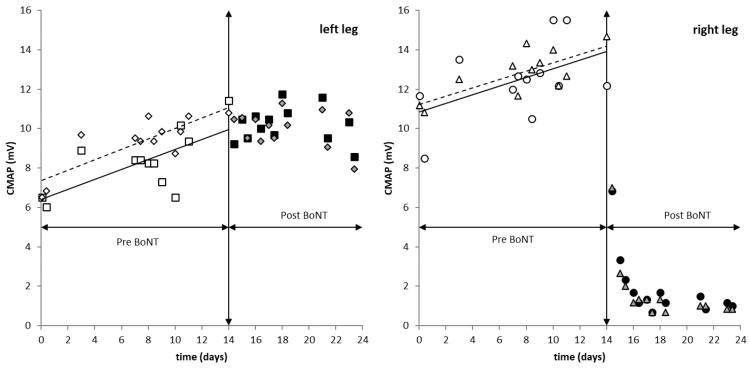
Development of the left EDB CMAP (**left side**) and the right EDB CMAP (**right side**) during 14 days of VET (Pre BoNT) and 10 days after injection of 20 U incoBoNT/A into the right EDB without VET (Post BoNT); (open squares): left EDB CMAP before BoNT before VET of 20 min duration; (open diamonds): left EDB CMAP before BoNT after VET of 20 min duration; (full squares): left EDB CMAP without VET before a pause of 20 min duration; (full diamonds): left EDB CMAP without VET after a pause of 20 min duration; (open circles): right EDB CMAP before BoNT before VET of 20 min duration; (open triangles): right EDB CMAP before BoNT after VET of 20 min duration; (full circles): right EDB CMAP without VET after injection of BoNT at day 14 before a pause of 20 min duration; (full triangles): right EDB CMAP without VET after injection of BoNT at day 14 after a pause of 20 min duration.

**Table 1 toxins-13-00423-t001:** Comparison of the left and right EDB CMAP before and after a 20 min vibration ergometry training (VET) over 14 days (day 0–14) and before and after a 20 min period of rest without VET over the following 10 days (day 14–24).

	Left EDB CMAP (mV) MV/S.D.	Left EDB CMAP (mV) MV/S.D.	Signif. Level *p*<	Right EDB CMAP (mV) MV/S.D.	Right EDB CMAP (mV) MV/S.D.	Signif. Level *p*<
	Before *n* = 12	After *n* = 13		Before *n* = 12	After *n* = 13	
day 0–14 VET	8.29/1.51	9.31/1.32	0.008	12.46/1.83	12.79/1.16	n.s.
day 14	no	BoNT injection	into the left EDB	injection of	20 U incoBoNT/A	into the right EDB
day 14–24 no VET	10.20/0.87	10.02/0.88	n.s.	1.90/1.58	1.68/1.62	0.02

**Table 2 toxins-13-00423-t002:** Increase in EDB CMAP (correlation regression line) before and after a 20 min vibration ergometry training over 14 days (day 0–14).

Parameter	Left EDB	Left EDB	Right EDB	Right EDB
	Before	After	Before	After
slope (mV/day)	0.2524	0.2643	0.2207	0.2114
intercept (mV)	6.429	7.359	10.83	11.23
r=	0.6805	0.8180	0.4902	0.5532
significance level *p*<	0.05	0.01	n.s.	0.01

EDB = extensor digitorum brevis muscle; CMAP = compound muscle action potential; MV = mean value; S.D. = standard deviation; *n* = number of measurements; r = coefficient of correlation.

## Data Availability

Data available on request due to restrictions eg privacy or ethical. The data presented in this study are available on request from the corresponding author.

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
