# Peer review of "Enhanced Effect of Botulinum Toxin A Injections into the Extensor Digitorum Brevis Muscle after Local Mechanical Leg Vibration: A Case Report"

_toxins, 2021, doi:10.3390/toxins13060423_

Round 1
Reviewer 1 Report
The authors sufficiently addressed all raised concerns, and also raised a few additional points that still should be addressed and are listed below. Overall, this study still presents a short report rather than a controlled study, and thus would be more suitable for a medical observation communication than a publication in a scientific Journal.
line 31: 600 kDa large complex proteins? Please correct or word more precisely.
line 45: Please mention whether citations 11 and 12 note or examined any potential effect of vibration on BoNT potency or duration for cosmetic applications.
Line 49: correct grammar
Table 1: There still is no mention of BoNT treatment in the table. The table should be understandable without having to skip back and forth through the text. Please indicate the injection of BoNT.
Lines 179-181: While the language is improved, the authors still do not show data that directly indicate growth of vessels and muscle fiber, and as such should not state this. Stating the the data are consistent with previous studies that also demonstrated vessel and muscle fiber growths would be an acceptable statement.
Lines 183-187: Thanks for adding this background information. However, the wording is very vague and should be much more precise. How much of an increase was observed, was it statistically significant, etc.
Line 204: change CAMP to CMAP
Line 194-205: Again, this line of argumentation is not scientifically precise. It is a correlation to other studies. Without direct comparison, no conclusion can be drawn as to whether VET increased BoNT potency and decreased onset time, as much as it seem probable.
Author Response
Reviewer 1 is right. The corresponding sentence is worded more precisely.
It is mentioned that in these studies vibration was used to reduce pain, but not for potential effect on BoNT potency.
Grammar has been corrected.
Information on BoNT is added in Table 1.
We now state that the results of the present study are consistent with previous studies demonstrating vessel and muscle fiber growth.
Reviewer 1 is right:
We have now added information that this increase was about 0.9mV in the mean and was significant (p<.05).
Has been corrected.
Reviewer 1 is right: We did not analyze EDB injections without VET.
We therefore emphasize in the conclusions that our results need confirmation by studies comparing the efficacy of BoNT injections with and without VET.
Reviewer 2 Report
All my suggestions from the previous revision have been included in the current version. I recommend it for publication.
Author Response
The authors are also thankful to reviewer 2.
Reviewer 3 Report
I wish to congratulate authors on such nice idea and very well performed study. Only minor suggestion for abstract - objective part. This sentence is not enough clearly written.
Author Response
The authors are also thankful to reviewer 3.
We have modified the abstract.
Round 2
Reviewer 1 Report
Authors addressed all questions.
This manuscript is a resubmission of an earlier submission. The following is a list of the peer review reports and author responses from that submission.
Round 1
Reviewer 1 Report
This manuscript presents a case report of a single patient. The authors report a very interesting finding that onset of paralysis and time to peak after BoNT treatment was significantly shortened after stimulation of neuromuscular junction activity by VET prior to treatment. However, this finding is based on comparison of this results from this single patient with a history of injury to previously reported results from studies using healthy volunteers, limiting the ability to interpret the results.
Overall, this case report, while interesting, does not rise to the level of a scientific report, as it lacks a comprehensive literature background and discussion. It is also somewhat difficult to read, as the goal of the study is not clearly focused and the English language needs significant correction.
Specific comments are listed below:
Lines 19-21: The conclusion is a bit unclear. Did previous studies also inject with BoNT? Please reword to be more precise.
Line 34: replace continuous with repeated or periodic
Table 1: This table is a bit confusing. What exactly do the numbers represent, in particular the numbers after the slashes? The time periods should contain information on treatment, especially whether VET was done and the BoNT injection. The methods state statistical analysis. Please clarify in the table the n number and whether average and standard deviation or error is shown.
Lines 167-168: ‘As soon as VET was stopped’: This is very confusing here, as the methods state that after day 14 VET was not conducted anymore.
Lines 179-181: if an increase of muscle was suspected, was the muscle mass measured in some way? Same for blood vessels.
Lines 203-211: The comparison to previously conducted studies some of which are using different toxin formulations and doses, and possibly different injection techniques, is uncontrolled and not a valid comparison. In addition, the present case report describes BoNT treatment of a previously injured and weakened leg, which further casts doubt on the comparability to previous data using healthy volunteers. It is fair to base a hypothesis on the presented data given previous literature reports, but it should be emphasized that a controlled study is required to examine this hypothesis further.
Lines 218-224: The conclusions were a surprise to me after reading the manuscript. They’re well stated and reasonable, however, it was unclear that this was the goal of the study. Also, why are the conclusions limited to this one aspect of BoNT treatment?
General Comment: There is extensive literature on the role of synaptic activity in BoNT uptake. A comprehensive literature review should be conducted and this literature from cell based and, if available, animal studies should be included in background or discussion.
Reviewer 2 Report
This is a very interesting case report which can be treated as a pilot study, but in my opinion, using the term “pilot study” should be dedicated to more numerous study group. I also have some other comments:
Many editorial bugs, please carefully check the manuscript according to that issue
Maybe the authors have some scheme of prototype vibration ergometer. It will be very interesting to demonstrate it with more details
Too laconic description of the study subject, please add full clinical characteristic
The number of bioethics commission must be provided